# Investigation of Scaling and Inhibition Mechanisms in Reverse Osmosis Spiral Wound Elements

**DOI:** 10.3390/membranes12090852

**Published:** 2022-08-31

**Authors:** Alexei Pervov

**Affiliations:** Department of Water Supply and Sewage, Moscow State University of Civil Engineering, 26, Yaroslaskoye Highway, 129337 Moscow, Russia; ale-pervov@yandex.ru

**Keywords:** reverse osmosis, calcium carbonate, scaling mechanism, antiscalants, nucleation, crystallization mechanisms

## Abstract

Understanding of crystal formation and growth conditions in reverse osmosis membrane channels enables us to develop efficient tools to control scaling in membrane facilities and increase their recoveries. Crystals are formed in “dead areas” and subsequently get out of them and sediment on membrane surface. Adsorption of polymeric inhibitor molecules to crystal surface was investigated as well as antiscalant behaviour throughout nucleation in “dead areas” and growth of crystals sedimented on membrane surface. Experimental dependencies of antiscalant adsorption rates on the antiscalant dosage values were determined. Examination of SEM images of crystals demonstrated that their size and amount depend on the supersaturation value reached in the “dead areas”. More efficient antiscalants delay the beginning of nucleation and reduce the rate of crystal growth due to adsorption and blockage of crystal growth process. Antiscaling property of inhibitors is also attributed to their ability to provide certain amount of adsorbent to block crystal growth during nucleation. A test procedure is described that enables us to predict concentrate composition in the “dead areas” and calculate supersaturation values that correspond to beginning of nucleation.

## 1. Introduction

Reverse Osmosis (RO) membrane water treatment technique is widely applied in the field of drinking water supply to remove hardness, fluoride, strontium, ammonia, lithium, boron and other ionic species to meet WHO standards. Meanwhile, successful growth of RO application is hindered by operational problems such as: concentrate handling and high reagent consumption. Both problems are related to scale formation in membrane modules [1,2]. This problem is partly solved through the use of antiscalants. But for specialists who operate RO facilities following questions are always relevant: which brand’s product is more effective and what are the requirements for the product purchased based on the results of the tender. Until now, there is no understanding among specialists how to evaluate antiscaling properties of the product. Express methods based on seed crystals growth or on observation of the beginning of nucleation often do not reflect real conditions that exist in membrane modules during scale formation. This article summarizes more than forty years of experience in studying the problem of sparingly soluble salts deposition on reverse osmosis membranes, development of scale prevention methods as well as formation of views on the mechanism of this process.

Examination of membrane surface after dead scaled-up spiral wound membrane autopsies contributed to the introduction of a new term into reverse osmosis research practice—“dead areas”. “Dead” or stagnant areas occur due to uneven flow distribution and uneven flow velocities in membrane channel. Flow velocities in “the dead area” are lower than in the main stream in membrane channel. Therefore, inside this area above membrane surface concentration polarization occurs and subsequently salt concentration values within this area are higher than in the main flow above membrane. The dead area in membrane separation process is recognized as the membrane area with low cross-flow velocity of the separated solution, which causes an increase of concentration polarization and leads to a decrease of rejection and increase of permeate salinity. The increase of concentration polarization also leads to the increase of salt concentration at the membrane surface that causes supersaturation of sparingly soluble salts and their precipitation. The level of concentration polarization does not grow indefinitely. The volume of liquid above the dead area of membrane surface follows the mass balance: the amount of salts that enters the dead area from the bulk solution is equal to the amount of salts withdrawn with permeate. Nucleation exactly occurs in dead areas due to concentration polarization. The maximum possible salt concentration values reached in “dead” areas can be calculated basing on salt balance considerations. The salt balance conditions in dead area correspond obviously to equilibrium state: the amount of salts that enter the dead area equal to amount of salts that penetrate through membrane. To reach in permeate salt concentration value that equals to salt concentration in the feed water (in the main solution), salt concentration value in the dead area on membrane surface should be determined from the equation:(1)$$C_{d}=C_{f}\times 100/R$$
where: *C_d_* is salt concentration in the dead area, *C_f_* is salt concentration in the feed water and *R* is membrane rejection, %.

But to describe mechanism of calcium carbonate and calcium sulphate crystal formation, values of supersaturation reached in the “dead areas” are still unknown.

According to crystallization theory [3], the first phase (homogeneous nucleation) occurs after supersaturation is reached. Nucleation rate (number of nuclei formed per unit of time) depends on supersaturation rate [4]. The higher supersaturation is, the more nuclei are formed and the smaller is their size. The smaller size crystals have in deposits, the higher was the supersaturation when they were formed. After nuclei are formed they start growing in supersaturated solution. Dependence of crystal growth rate on supersaturation value usually is described by the formula proposed by Kimura and Okazaki [3] and also by J. Gilron and D. Hasson [4], where the rate of crystal growth is proportional to the square of the supersaturation value:(2)$$dM/dt=\left(C_{b}-C_{s}\right)2\times K$$
where *M* is the specific mass of deposited calcium carbonate on membrane area;

*C_b_* is saturation concentration;*C_s_* is concentration of salt in solution;*K* is constant.

Crystals are formed in the “dead” area and are carried out from it by the turbulent flow, subsequently sediment on membrane surface and continue to grow in supersaturated solution. Therefore the processes of crystal formation and their growth occur in different conditions. The mechanism of removal of crystals from “dead” areas is still not developed. But explanation of this process could be found in the articles of Georges Belfort, who developed the “migration” theory based on the assumption that particles with a larger mass are carried away faster by a turbulent flow the “dead” area [5].

In 1991 it was reported [6] that the dead areas are responsible for crystal nucleation are formed in the places where spacer mesh bungles are pressed to membrane surface (Figure 1).

Crystal nucleation is caused due to an increase in concentration by 10–20 times compared to the concentrations in the concentrate stream. Homogeneous nucleation occurs at the membrane surface and after that the formed crystals are carried out from the “dead” areas according to the particulate fouling mechanism. Figure 2 shows photos of membrane surface made after spiral wound element autopsies. Scaled-up “dead” areas and areas where crystals deposited on membrane surface are seen. Lately all membrane surface become covered by crystals due to removal of crystals from “dead” areas.

Figure 3 shows the dead area under the places where spacer mesh is pressed to membrane surface during winding, crystals formation and their withdrawal back into the flow.

The study of crystallization rates under different conditions has been studied by the author for 30 years [6]. Various approaches to control scaling in reverse osmosis desalination systems were studied [7,8,9,10], as well as the use of antiscalants [11,12,13,14] and descriptions of scaling and inhibiting mechanisms [15,16,17].

Scaling rates and antiscalant efficiencies are determined in commercial spiral wound modules and crystal formation is observed in real industrial conditions. The method of calcium carbonate scaling rate determining was developed on the basis of mass balance [6].

Comparison of determined scaling rate values enables us to compare efficiencies of measures applied to control scaling, particularly to compare inhibiting efficiencies of various antiscalants, as well as to determine their required doses and their influence on calcium carbonate deposition [15].

Application of antiscalants to control sparingly soluble salts scaling in reverse osmosis systems today became most important and significant in world water treatment practice [6,7,8,9,10,11,12,13,14,15,16,17,18,19,20].

As it was already discussed [7,8,9,10], antiscalant testing techniques used by different researchers to evaluate antiscaling efficiencies [8], often do not provide real conditions for crystal nucleation that exist in industrial spiral wound modules. This does not enable us to efficiently apply new antiscalants and evaluate their application behaviour [21,22,23,24,25]. A number of testing techniques are based on evaluation of growth rate of seed crystals added to the supersaturated solution in the presence of antiscalants [16,18]. In some cases antiscalant is added to supersaturated solution and antiscaling efficiency is determined by the supersaturation value that corresponds to visible nucleation beginning, that is controlled by jump of calcium concentration or colloid concentration [25]. Often laboratory testings provide inconsistent results that cannot be transfered into practice [11,12,13,14,15,16,17,18,19,20,21,22]. Also, different authors present their research results based on different understanding of scaling and inhibition mechanisms [26,27,28,29,30,31].

To understand nucleation conditions, concentration values of all species should be determined within the “dead area”. Previously conducted research [6] discovered, that the more efficient antiscalant is, the smaller is the size of the crystals formed in RO module. Nucleation theory [3,4] claims that nucleation rate which is determined as a number of nuclei formed in the unit time depends on supersaturation ratio: the higher supersaturation is, the more crystals are formed in a unit time. And also the higher supersaturation is, the smaller are crystals formed. Calculation of crystals number formed in the presence of various antiscalants under similar conditions revealed that the more efficient antiscalant is the higher supersaturation value corresponds to the beginning of nucleation. This made us to draw to an assumption that in the presence of antiscalants the beginning of nucleation is “observed” at higher supersaturation when antiscalant “fails” to “block” growth of crystal structure or in other words it’s inhibiting/adsorption abilities are unsufficient.

Rejection and permeability characteristics of membranes also influence the scaling propensities of membrane modules. It can be explained by the dead areas formation in on membrane surface and increase of concentration polarization ratios in these areas that tend to supersaturation and deposition of sparingly soluble salts. Rejection characteristics of membrane as well as their flux influence concentration polarization ratio and thus influence scaling rates. This was confirmed by a number of researches, where nanofiltration membrane spiral wound modules demonstrated lower scaling propensities than modules tailored with reverse osmosis membranes [7].

In previously published articles results of antiscalant adsorption rates were determined [32]. To determine antiscalant concentrations in water, a fluorescent method was developed and applied [33]. The use of “fluorescent label” also enabled us to make scan microscope and to “visualize” a process of antiscalant adsorption on crystal surface and on membrane surface [34]. It has been proven that antiscalant binds to calcium in water solution and does not adsorb on membrane surface. Adsorption of antiscalant on membrane surface was detected only if water did not contain calcium [35].

Thus, the goals of this research were:-to determine the composition of water in the “dead areas” and to evaluate supersaturation values that correspond to the start of nucleation in the presence of various antiscalants;-to determine rates of nucleation and dependencies of nucleation rates on supersaturation for different conditions and antiscalant doses;-to determine dependencies of calcium carbonate crystal growth rates on supersaturation in the presence of different antiscalants with different doses.

## 2. Experiments. Materials and Methods

The program included three series of experiments:Determination of calcium carbonate crystallization rates.Determination of antiscalant adsorption rates.Evaluation of supersaturation ratios reached in “dead areas”.Determination of crystal sizes depending on antiscalant.

The laboratory test unit flow diagram is shown on Figure 4. Principles of experiment conductance to determine calcium carbonate scaling rates and evaluate antiscalant efficiencies are described in [32]. Laboratory test unit was operated in circulation mode with constant permeate discharge. The feed water (Moscow tap water) was added to the feed water tank 1. From tank 1 the feed water was delivered by the pump 2 to membrane module 3. Spiral wound elements of 1812 standard (model 1812 R 100) tailored with low pressure RO BLN membrane were used. Membrane surface in 1812 module was 0.5 square meter. Retentate was returned back to the tank 1 while permeate was forwarded to permeate tank or discharged into the drain. The valve 9 regulated pressure value. Pressure value was maintained 7.0 Bars. The volume of circulated retentate was determined using the scale on the tank 1 wall. Retentate samples were withdrawn from tank 1 and permeate samples were taken from the outlet of permeate tube. Samples of circulating concentrate were withdrawn from tank 1, as well as permeate samples were taken from the hose that delivered permeate to tank 5. Temperature, electrical conductivity, pH, calcium, chloride, sulphate concentrations, alcalinity values were determined in the samples.

Electrical conductivity, TDS and temperature values were determined using laboratory conductivity meter model Cond 730 (WNW inoLab). pH values were determined using laboratory pH meter HI 2215 (Hanna Instruments) was determined using titration method, calcium concentrations were determined by EDTA titration.

The first and second series of experiments were conducted using Moscow tap water. TDS of tap water was 240–270 ppm; calcium concentration was 4.5 milliequivalents per liter; alkalinity was 100–110 ppm; chloride concentration was 25–30 ppm; sulphate ion concentration was 10–13 ppm; pH was 7.1.

In the process of experiments conducted in circulation mode the coefficient K values were determined that were initial volume in tank 1 in the beginning of experiment to values of concentrate volume at the certain moment when concentrate sample was withdrawn.

In the first series scaling rates in membrane modules were determined in the presence of different antiscalants at different doses. Scaling rates were evaluated using the method developed by the author [6,33,34,35]. The experimental procedure included determination of calcium concentration at different moments that corresponded to different volumes of concentrate in tank 1 and different coefficient K values. Figure 5 shows the dependencies of calcium ion concentration on coefficient K values in the presence on various antiscalants. Concentrations of phosphate ions were determined throughout each test run to evaluate antiscalant adsorption rates on the surface of formed crystals. Figure 6 shows dependencies of antiscalant concentration on K throughout each test run.

In the second series ionic compositions of permeate and concentrate were determined throughout circulation experiments and growth of concentration K values, where K reached values of 30 and 40. To predict TDS value and concentration of different water species in the “dead” area permeate TDS should reach TDS value of the feed water that enters this “dead” area (Figure 1). Therefore, we increased K value until product water TDS equaled feed water TDS. Figures show the obtained dependencies of TDS, calcium, chlorides, alkalinity, sulfates, as well as antiscalant concentration values in permeate and concentrate throughout the test run on the concentration coefficient K and TDS values. To determine TDS value in “dead” area in membrane module when K is 1.5, we have to determine concentrate TDS for the K = 1.5 on the graph shown on the Figure 7. Then, using the graph shown on Figure 8 and using the obtained TDS value of permeate, we can determine concentrate TDS that corresponds to the required permeate TDS. This value is TDS value in the “dead” area. Similarly, using graphs presented on Figure 7 and Figure 8, we can determine concentrations of bicarbonates, chlorides and calcium ions in “dead” areas that enable us to evaluate supersaturation and nucleation conditions.

The third experimental series included scale samples preparation and making photos using scanning electron microscope (SEM). By the end of each test run membrane flushing procedure was arranged whereby concentrate flow and pressure regulation valve 10 (Figure 4) was opened that caused pressure drop and rapid increase of transit flow through membrane module. The formed crystals were thus flushed off membrane surface and swept with concentrate flow into the tank 11. The flushed concentrate was then filtered through membrane microfilter MFAS-OS-3, the sediments accumulated on membrane filter surface were rinzed with distilled water and dried at temperature of 50 degrees (Celsium). Then crystal deposits were transfered to scotch and examined using SEM methods. Scanning electron microscope with thermoemission cathode Quanta 250 FEI Company and then the energy dispersion (GENESIS APEX 2 EDS System with APOLLO X SDD EDAX) was used. Observations were performed using potential of 12.5 and 15 Kilovolt at low vacuum mode. The further analysis of crystal and experimental data processing enabled us to elaborate dependencies of nucleation and crystal growth on supersaturation and evaluate the influence of antiscalants.

The conducted experiments enabled us:-to detect supersaturation conditions and reveal the role of antiscalants in inhibiting nucleation and crystal growth rates;-to understand influence of inhibitor chemical composition on inhibiting activity;-to determine antiscalant adsorption rates during nucleation and crystal growth phases.

## 3. Discussion of Results. Processing of the Experimental Data

Figure 9 shows main steps to determine calcium carbonate scaling rates in the presence of antisalants and without antiscalant addition. The amount of calcium carbonate was determined by a mass balance as a difference between the amount of calcium in tank 1 in the beginning of the experiment and calcium amount in tank 1 at the certain moment of experiment [6]. Figure 9a demonstrates dependencies of deposited calcium carbonate amounts on coefficient K values. Calcium carbonate deposition rates were determined in conformity with the method described in [6] as tangents of the function of calcium carbonate amount versus time (Figure 9b). Results of scaling rates determination are demonstrated on Figure 9c as dependencies of calcium carbonate scaling rate values (expressed as milliequivalents of calcium carbonate per hour) on coefficient K value. Figure 9c demonstrates coefficient K values that correspond to beginning of scaling in RO module under different conditions in the presence of antiscalants and without antiscalant addition. Different K values correspond to different scale formation (nucleation) conditions in the “dead” area.

The number of previous research showed that “Aminat -K” was recognized as the main efficient home-made antiscalant [33,35]. The choice of other phosphonic acid-based antiscalants that are weaker than “Aminat-K” (Actoscale and Jurby-soft) for comparison can be explained by the presence of these products at Russia desalination market [33]. As it can be seen at Figure 9, AMINAT demonstrates lower scaling rates than other antiscalants at different doses. Figure 9c shows dependencies of accumulated amount of calcium carbonate on time. As it can be seen, at different antiscalant doses the beginning of calcium carbonate accumulation (the K value when calcium carbonate amount is zero) corresponds to different K values and to different supersaturation values in the “dead” areas respectively. Scaling rates were determined as tangents values of M versus K graph (Figure 9b). When scaling rate was determined, author assumed that this value K on abscissa (when mass of scale is zero) corresponds to beginning of crystal formation. Thus, the experimentally obtained scaling rate values correspond to the scaling rate values of the first phase of scaling—homogeneous nucleation. Figure 9 shows that the more efficient antiscalant is used, the lower is nucleation rate and the smaller are crystals formed. Also this indicates that the more efficient antiscalant is used, the higher supersaturation is required to initiate nucleation. Supersaturation conditions in the “dead” areas that correspond to various K values can be determined using Figure 7 and Figure 8.

The previously conducted investigations showed that with calcium carbonate rate growth also antiscalant adsorption rate increases [35]. Figure 10 shows results of determination of adsorption of Aminat-K to crystal surface. Adsorption rates were determined similarly to scaling rate determination (Figure 9). During calcium carbonate rate determination concentrations of antiscalant (phosphate-ion) were determined and amount of accumulated antiscalant as well as rates of its accumulation (adsorption) were determined (Figure 10). As it can be seen on Figure 10, the increase of antiscalant doses causes the growth of its adsorption rate to crystal surface. Antiscalant adsorption rate at the initial moment of scale growth (Figure 9) also corresponds to adsorption rate during nucleation phase.

The obtained described results offer main steps of scaling mechanism description in spiral wound membrane modules. Homogeneous nucleation occurs in “dead” areas after the certain supersaturation value is reached. Each supersaturation value corresponds to the certain value of nucleation rate that is a number of nuclei formed in the unit volume in a second. Author drew to the conclusion that this nucleation rate value remains the same throughout all experimental tests run even at different K values and different supersaturation values. This is proven by the photos of crystals withdrawn from the module in the end of each test run [34] that have the even size distribution (Figure 11). As it is shown on Figure 11, the higher antiscalant dosage is, the smaller is the size of crystals formed. In other way, if various supersaturation conditions occurred in “dead” areas throughout the test run, crystals of different size would have been formed. The higher supersaturation value is, the smaller crystals are formed and the higher is a number of crystals. Moreover, the dependency of scale mass on time (Figure 9,c) would not be linear but have constant increase as the increase of supersaturation in the “dead” areas could have caused the growth of a number of formed nuclei and rapid growth of amount of accumulated calcium carbonate.

The claimed hypothesis can have the following explanation:Nucleation occurs in the “dead” areas after the certain supersaturation value is reached.After this supersaturation, value is reached nucleation phase starts and nucleation rate remains constant throughout experimental test run irrespectively on the K values and supersaturation values reached.

This thesis can be explained by the fact that nucleation starts after supersaturation reaches the certain value in the “dead” area. Nuclei are formed and can grow in the supersaturated solution. After supersaturation value is increased (with K value growth), the excessive calcium and carbonate ions participate in nuclei crystal growth. New nuclei are not formed.

It seems interesting to build dependencies of scaling rates on K, scaling rates on supersaturation values for different antiscalant doses, nucleation rates on supersaturation. These relationships can help to understand better the mechanism of scale formation and its control using antiscalants [32,33].

To determine nucleation rates we used SEM photos of crystals withdrawn from the modules after operation without antiscalant addition and after dosing 2, 5 and 10 ppm of “Aminat-K”. Using the size of crystals, to simplify calculations, assuming cubic shape of crystals, the volume and mass of one crystal were calculated (Table 1 and Table 2).

Having determined the total mass of accumulated calcium carbonate during the test run, the amounts of crystals formed were calculated. Crystal formation rate depends on supersaturation in the “dead” area. This supersaturation value that corresponds to the beginning of nucleation can be determined using Figure 7 and Figure 8 and corresponds to certain value of concentration coefficient K.

Figure 12 shows the calculated nucleation rates N depending on supersaturation values. Calculation of supersaturation values in concentrates and “dead” areas was conducted in conformity with the well-known method described in [34,36]. Results of calculations are presented in Table 1. Results of calculation of crystal size, mass and other parameters are presented in Table 2. Supersaturation value was calculated as a ratio of calcium and carbonate ionic concentrations product to the value of calcium carbonate solubility product Ksp. Carbonate ions concentration were determined from the value of S determined as a ratio of carbonate ions concentration to bicarbonate ions concentration:(3)$$S={CO}_{3}/{HCO}_{3}$$

The *S* value was determined using graphs presented in [36]. Nucleation rate values were calculated for different cases of crystal formation in the presence of antiscalants at various *K* values, identified as *K_f_* (Figure 12).

Results of calculations are shown on Figure 12 and Figure 13 as dependencies of N on K and on supersaturation.

It is also interesting to describe dependencies of antiscalant adsorption rates on the nuclei surface formed in the beginning of nucleation phase. Figure 13 shows results of Aminat-K and Jurbysoft adsorption rates determination depending on the crystal surface that corresponded to calculated values of N. As it can be seen on Figure 13a, Aminat-K ensures higher adsorption rates than Jurby-Soft even at the same doses. For different K values a relationship between adsorption rates and antiscalant doses were built (Figure 13b). Figure 13 shows that various antiscalants demonstrate different adsorption abilities on various areas of crystal surfaces. More strong Aminat-K with 10 ppm dose provides antiscalant adsorption on crystal surface formed at K = 1.5, but Jurbysoft even with 10 ppm dose cannot provide such adsorption rate. Thus, using an approach demonstrated on Figure 13, we can compare antiscalant properties: first determine values of N and supersaturation value that corresponds to beginning of nucleation at certain K value; determine antiscalant adsorption rate at this K value; determine what antiscalant does is required to ensure this adsorption rate. Figure 13 also explains why Aminat-K provides same scaling rate values both with 2 and 7 ppm doses [33,34,35] and Jurbysoft demonstrates different values of scaling rates with the same doses. Aminat-K is a strong antiscalant and even at small dose of 2 ppm provides higher than Jurbysoft adsorption rates to inhibit crystal nucleation.

The obtained relationships (Figure 12 and Figure 13) show the total final description of the process of crystal formation and growth on membrane surface. As it was already claimed [6,33], crystal nuclei form in the “dead” areas, get out of them and subsequently sediment on membrane surface. During reverse osmosis module operation the processes of crystal nucleation and crystal growth on membrane surface occur simultaneously. This significantly complicates separate description of these processes as well as investigation of the influence of antiscalants on these processes.

To provide a nucleation process description, an assumption was made that we can divide the process in periods that correspond to increase of K value within limits: K changing from 1 to 2; K changing from 2 to 3; K changing from 3 to 4; K changing from 4 to 5. Then we can determine the amount of calcium carbonate, formed during the period when K changed from 2 to 3, from 3 to 4 and from 4 to 5. Then we have to subtract from this mass value the amounts of calcium carbonate formed in the “dead” areas during time when various K values were reached. These values were calculated, assuming determined nucleation rate value in the beginning of the scaling process (Figure 9,c). For each period, this value of N was multiplied by the time value required to reach K increase from 2 to 3, from 3 to 4 and from 4 to 5. Figure 14a shows results of calculated amounts of calcium carbonate during various periods of K value growth and resulting curve that shows dependency of amount of calcium carbonate increased due to growth of crystals on membrane surface on time. Dependencies of grown calcium carbonate on time and on supersaturation value are presented on Figure 14b,c.

To describe inhibition mechanism during crystal growth phase, we have to determine antiscalant adsorption rates depending on supersaturation and crystal growth values. Dependencies of antiscalant adsorption rates on supersaturation and scaling rate values (Figure 15) are obtained similarly to description of calcium carbonate crystals growth rate on membrane surface (Figure 13 and Figure 14). Experimental data processing provided values of antiscalant adsorption as a function of K, supersaturation, antiscalant dose and growing crystals surface. Figure 15 shows main steps of dependencies building: antiscalant adsorption rate on nuclei crystals on K value (a); amount of adsorbed antiscalant on K value (b); amount of adsorbed antiscalant on time (c); adsorption rate on K value; adsorption rate on crystal growth rate (d); adsorption rate on crystal surface area (e). All plots are built for the case of Aminat-K use with 10 ppm dose [35].

Basing on described results, mechanism of antiscalant behaviour in the “dead” areas can be explained. The more strong and efficient antiscalant is used, the higher supersaturation value is required and higher K value should be reached to begin scaling. Higher supersaturation values correspond to higher nucleation rate (number of nuclei formed in the unit volume in a second). The presence of antiscalant in supersaturation solution blocks/retards nucleation process—formation of noticeable crystals. Concentration of antiscalant in the beginning of nucleation plays an important role: the amount of antiscalant is enough to block small numbers of nuclei formed at small supersaturation value. At higher supersaturation values high number of nuclei is formed and the amount of antiscalant and its adsorption ability could be not enough to block nuclei growth. Figure 12 shows dependencies of scaling rates on K values. The beginning of scaling (nucleation phase) corresponds to the certain K value. Therefore, nucleation rate N is presented on Figure 16 as a function of supersaturation. The stronger antiscalant is, the higher is N and supersaturation value.

## 4. Industrial Application of the Results

Pilot testing program was arranged at “Shekino Azot” where 100 cubic meter per hour RO membrane unit was operated. Throughout Aminat-K dosing testing period cleanings were performed only once four months (compared to twice a four months using Jurby-Soft). Figure 17 shows the observed permeate flow reduction and delta-pressure increase during operational periods when Aminat-K and Jurby-Soft were dosed. Dosing amount for both antiscalants were 5 ppm.

Antiscaling efficiency provides a certain reduction of scaling rates and provides longer operational period between cleanings (Figure 18). Figure 18a shows results of scaling rates determination as a function on K for three commercial antiscalants (Aminat-K, Jurby-Soft and Actoscale) tested with 5 ppm dose using Moscow tap water (Figure 18a). Figure 18b shows the dependency of accumulated calcium carbonate amount (milliequivalents) on time, obtained earlier [32] for 1812 elements with RO low pressure membranes (BLN model) operated at Moscow tap water with Aminat-K addition dose 5 ppm. Similar plots are used for prediction techniques used in membrane facilities operation [6,32]. Figure 18b demonstrates the recommended operational time value of 300 hours that corresponds to calcium carbonate scale amount of milliequivalents. The recommended amount was chosen as a result of optimization investigation of chemical cleanings. For the cases when other antiscalants are used, that provide smaller efficiencies and higher scaling rate values, operational time required to reach cleaning should be shortened (Figure 18c). If we arrange cleaning procedure later, than it was recommended, the amount of accumulated scale grows and can the scale dissolution ability of cleaning procedure. Thus, certain amount of scale is lasted after non-efficient cleanings and increase. This explains why none successfully predicted values of operational time and selected antiscalants can cause constant permeate flux and rejection reduction during operation.

Phosphonates, antiscalants based on phosponic acids, are recognized as main efficient class of antiscalants. Yet, experimental and operational experience showed that various phosphonates demonstrate different antiscaling abilities. The difference in their behaviour is attributed to concentration of MEDF produced during the synthesis. Figure 19 shows the spectrums that provide data on concentration of MEDF. “Jurby-Soft” is a mixture of NTF and HEDP and does not contain MEDF (Figure 19a) therefore demonstrating the lowest antiscaling efficiency. Both “Aminat-K” and “Actoscale” are considered a mixture of NTF and MEDF, but MEDF to NTF concentration ratio in “Aminat-K” is higher than in “Actoscale” (Figure 19b,c). It is recommended to undertake control of the samples and to control after purchase to escape false (Table 3).

## 5. Conclusions

1. Crystal formation begins in the “dead” areas at the constant supersaturation value irrespectively of calcium and carbonate ion concentrations in RO concentrate.

2. The presence of antiscalant increases the supersaturation value that corresponds to nucleation beginning. Nuclei crystals are formed, and their number and size are dependent on the supersaturation value. The higher supersaturation is reached, the smaller crystals are formed and the higher is their number.

3. For scale inhibition process the ratio of aniscalant adsorption rate and crystal surface area are very important. Various antiscalants provide different adsorption rate values to the equal surface area at the equal supersaturation values. Thus, more efficient and strong antiscalants provide more adsorption material to the crystal surface enough to block crystal growth. When weak antiscalants are used their adsorption ability turns to be not enough to block the growth of formed nuclei.

4. Presence of antiscalants retards process of nucleation in the “dead” area. At low supersaturation levels and low nucleation rate values the growth of nuclei crystals is blocked by antiscalant. At higher supersaturation values, larger number of nuclei is formed and antiscalant adsorption ability could be not enough to adsorb on the large crystal surface area and to block their growth. Thus, process of nucleation and crystal growth begins in the “dead” area.

## Figures and Tables

**Figure 1 membranes-12-00852-f001:**
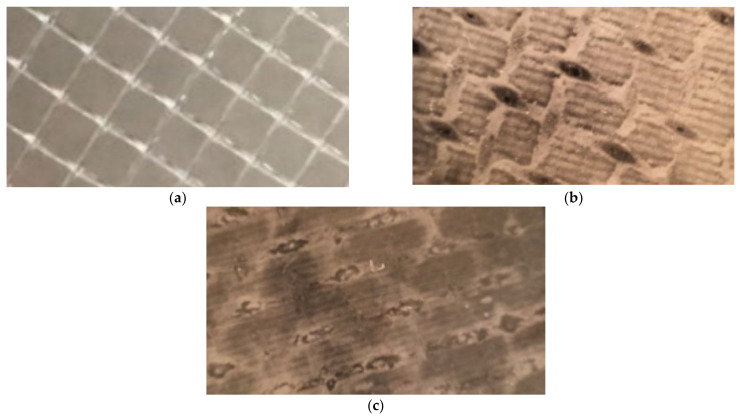
Formation of “dead areas” on membrane surface in spiral wound elements: (**a**) spacer mesh; (**b**) membrane surface in the beginning of scaling; (**c**) membrane surface covered by calcium carbonate scale layer.

**Figure 2 membranes-12-00852-f002:**
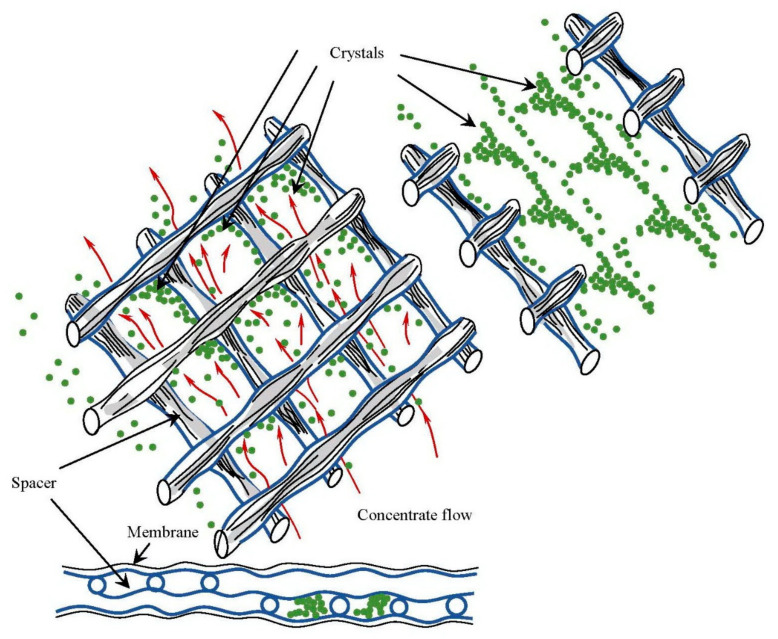
Formation of “dead area” in the places where the mesh touches the membrane surface.

**Figure 3 membranes-12-00852-f003:**
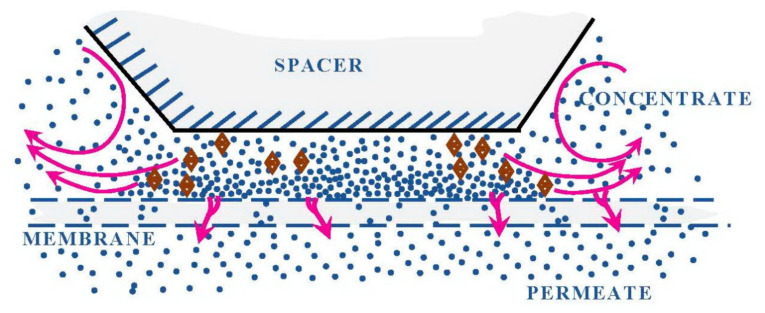
“Dead areas” formation on membrane surface due to spacer contact with membrane surface. Crystal formation and withdrawal from “dead areas” and further fouling of membrane surface.

**Figure 4 membranes-12-00852-f004:**
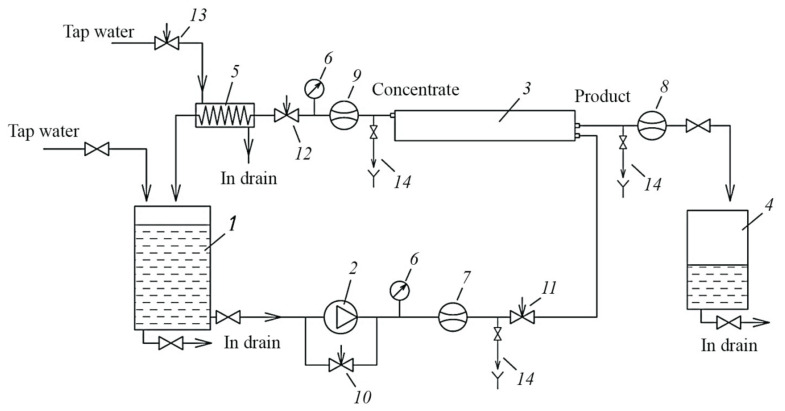
Schematic diagram of laboratory RO unit for membrane scaling tests: 1—feed water tank; 2—pump; 3—spiral wound membrane module; 4—permeate tank; 5—heat exchanger; 6—pressure gauge; 7—feed water flow meter; 8—permeate flow meter; 9—concentrate flow meter; 10—by-pass adjusting valve; 11—feed water adjusting valve; 12—concentrate adjusting valve; 13—cooling water adjusting valve; 14—sampler.

**Figure 5 membranes-12-00852-f005:**
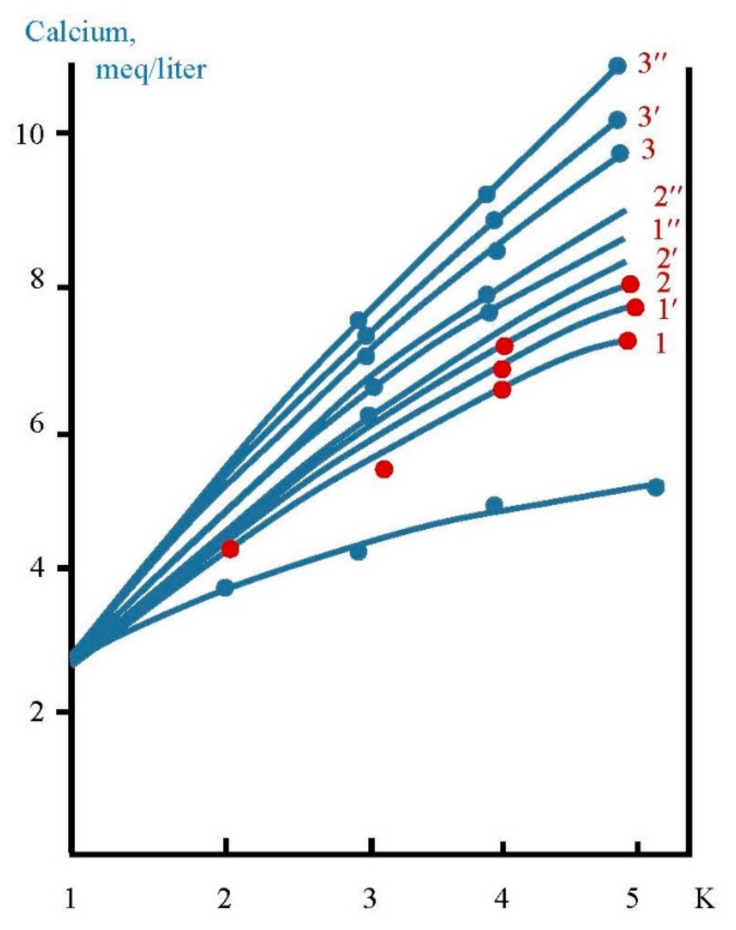
Dependencies of calcium ion concentration values on K during experiments: 1—Jurbysoft, 2 ppm; 1′—Jurbysoft, 5 ppm; 1′’—Jurbysoft, 10 ppm; 2—Actoscale, 2 ppm; 2′—Actoscale, 5 ppm; 2′’—Actoscale, 10 ppm; 3—Aminat-K, 2 ppm; 3′—Aminat-K, 5 ppm; 3′’—Aminat-K, 10 ppm.

**Figure 6 membranes-12-00852-f006:**
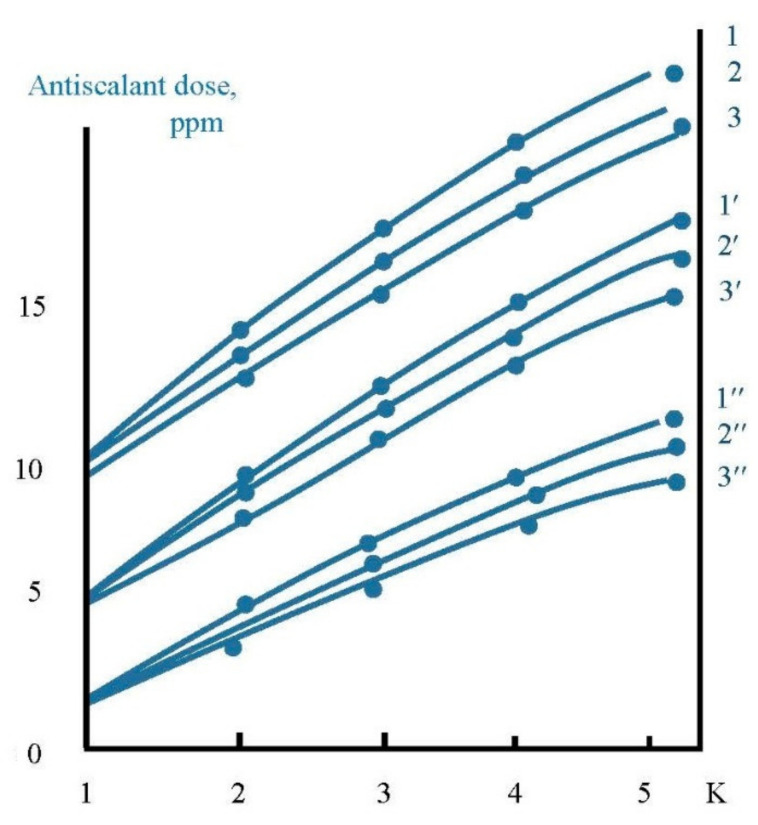
Dependencies of antiscalant concentrations on K during experiments: 1—Jurbysoft, 10 ppm; 2—Actoscale, 10 ppm; 3—Aminat-K, 10 ppm; 1′—Jurbysoft, 5 ppm; 2′—Actoscale, 5 ppm; 2′’—Actoscale, 2 ppm; 3—Jurbysoft, 2 ppm; 3′—Actoscale, 2 ppm; 3′’—Aminat-K, 2 ppm.

**Figure 7 membranes-12-00852-f007:**
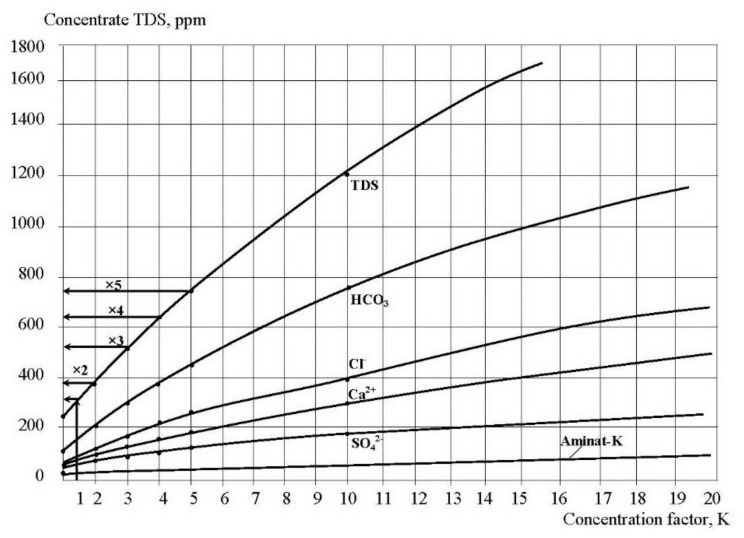
Dependencies of concentrate TDS and other species concentrations on K.

**Figure 8 membranes-12-00852-f008:**
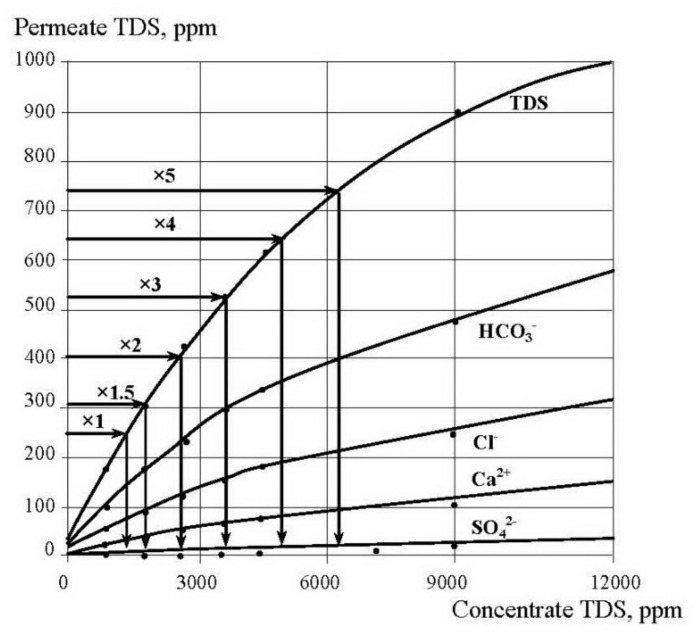
Dependencies of permeate TDS and concentrations of different species in concentrate on concentrate TDS.

**Figure 9 membranes-12-00852-f009:**
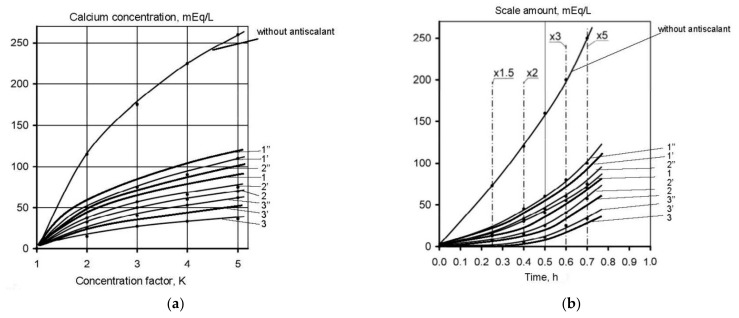
Determination of calcium carbonate growth rates on membranes in the presence of various antiscalants in different doses: (**a**) dependence of accumulated scale amount on K; (**b**) dependence of accumulated scale amount on K; (**c**) dependence of scaling rates on K. 1—Aminat-K, 2 ppm; 1′—Actoscale, 2 ppm; 1′’—Jurbysoft, 2 ppm; 2—Aminat-K, 5 ppm; 2′—Actoscale, 5 ppm; 2′’—Jurbysoft, 5 ppm; 3—Aminat-K, 10 ppm; 3′—Actoscale, 10 ppm; 3′’—Jurbysoft, 10 ppm.

**Figure 10 membranes-12-00852-f010:**
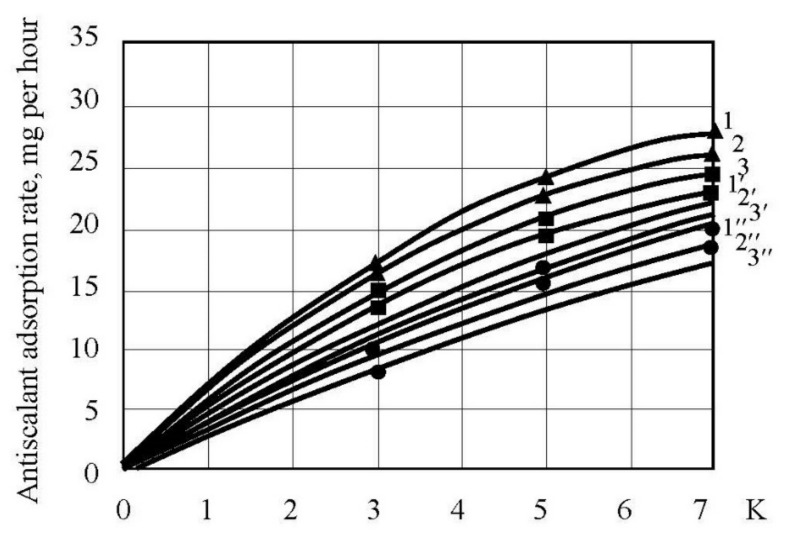
Results of evaluation of antiscalant adsorption rates: dependencies of adsorption rates on K. 1—Aminat-K, 10 ppm; 2—Aminat-K, 5 ppm; 3—Aminat-K, 2 ppm, 1′—Actoscale, 10 ppm; 2′—Actoscale, 5 ppm; 2′’—Actoscale, 2 ppm; 3—Jurbysoft, 10ppm; 3′—Jurbysoft, 5 ppm; 3′’—Jurbysoft, 2 ppm.

**Figure 11 membranes-12-00852-f011:**
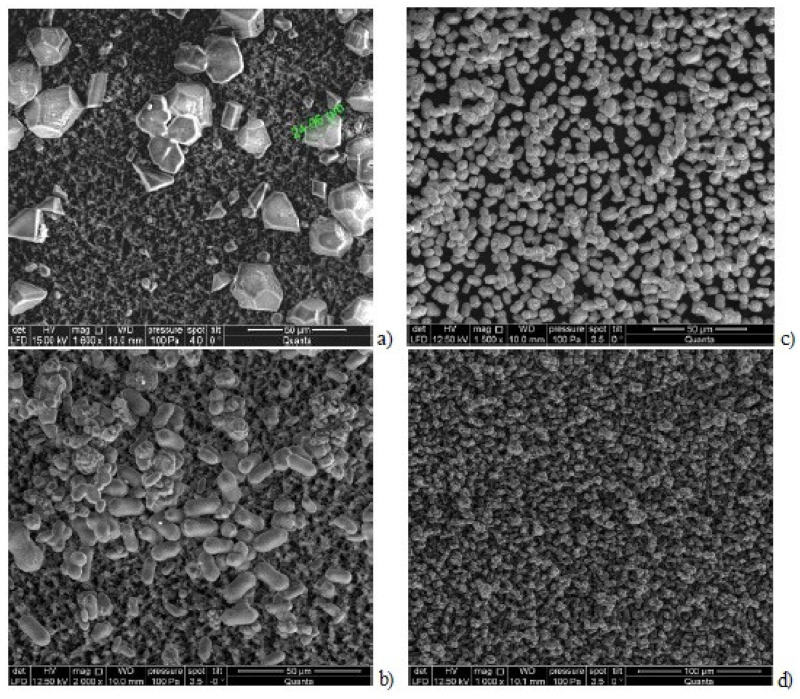
SEM photos of crystals withdrawn from membrane module: (**a**) without antiscalant addition; (**b**) Jurbysoft, 5 ppm; (**c**) Actoscale, 5 ppm; (**d**) Aminat-K, 10 ppm.

**Figure 12 membranes-12-00852-f012:**
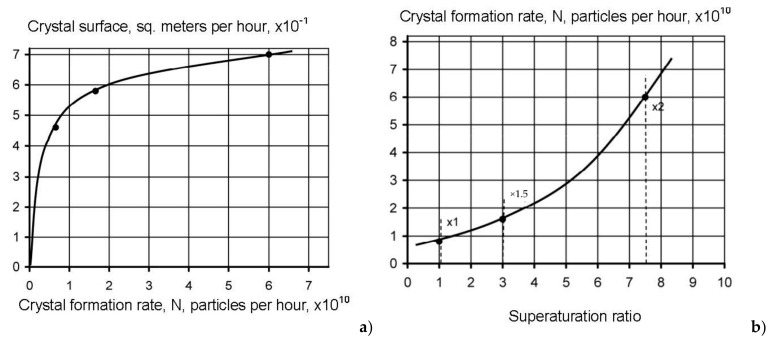
Dependencies of crystal surface area on nucleation rate N (**a**) and nucleation rate N on supersaturation (**b**) for the case of Aminat-K application: 1—Aminat-K dose—2 ppm, *K_f_* = 1.2; 2—Aminat-K dose—5 ppm, *K_f_* = 1.5; 3—Aminat-K dose—10 ppm, *K_f_* = 1.8.

**Figure 13 membranes-12-00852-f013:**
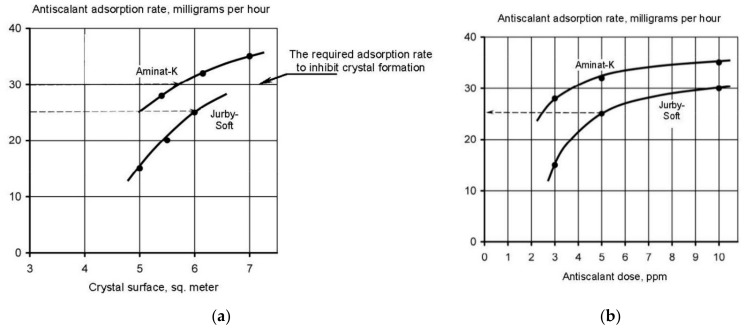
Results of Aminat-K and Jurbysoft adsorption properties evaluation: (**a**) dependence of adsorption rate on antiscalant dose; (**b**) dependence of adsorption rate on K.

**Figure 14 membranes-12-00852-f014:**
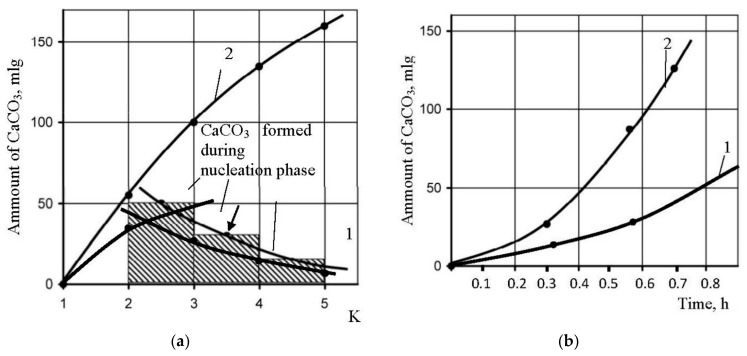
Evaluation of crystal growth rates: (**a**) dependencies of amount of grown calcium carbonate scale on K and amount of formed calcium carbonate during nucleation; (**b**) dependencies of grown calcium carbonate on time; (**c**) dependencies of grown calcium carbonate on supersaturation value.

**Figure 15 membranes-12-00852-f015:**
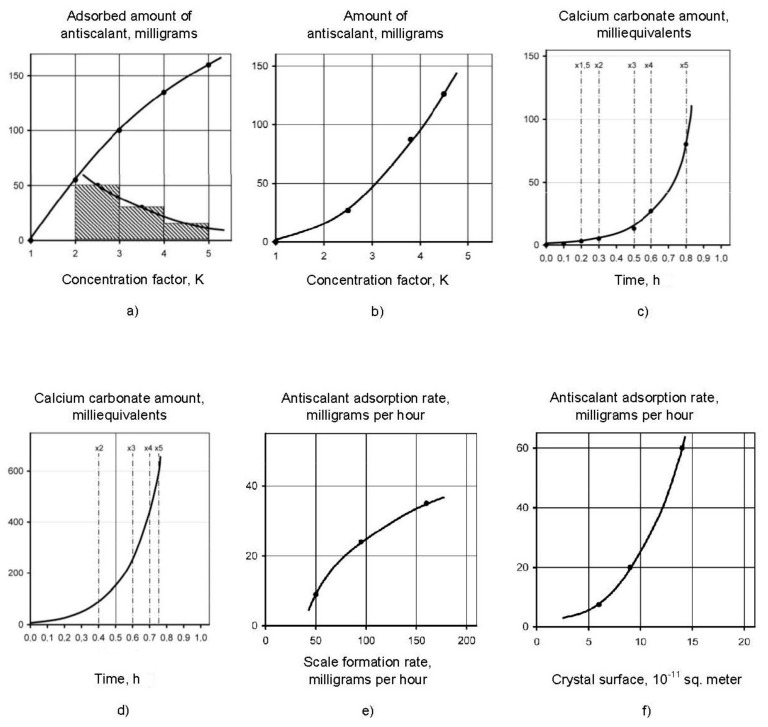
Steps to determine dependencies of: adsorption rate on K (**a**), amount of adsorbed antiscalant on K (**b**); amount of adsorbed antiscalant on time (**c**), antiscalant adsorption rate on time (**d**); antiscalant adsorption rate on crystal growth rate (**e**); antiscalant adsorption rate on crystal growth area (**f**).

**Figure 16 membranes-12-00852-f016:**
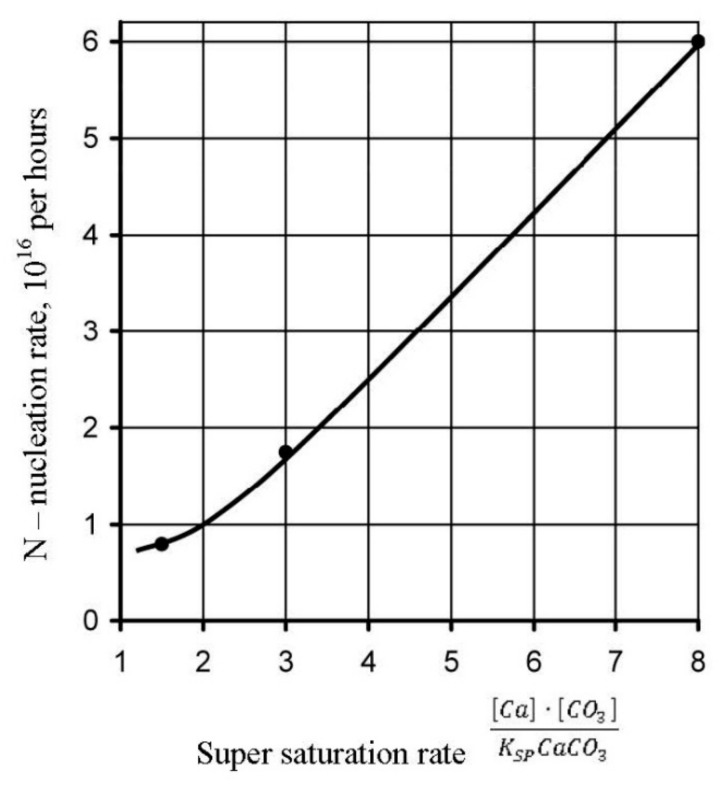
Dependence of nucleation rate N (number of nuclei formed per second) on calcium carbonate supersaturation ratio.

**Figure 17 membranes-12-00852-f017:**
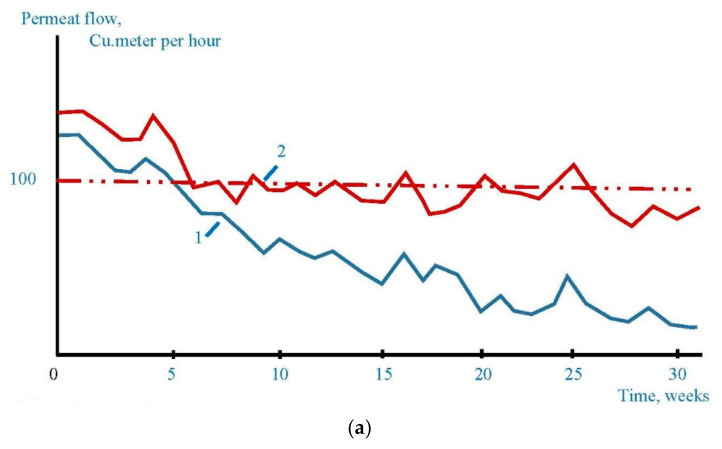
Comparison of the RO unit operation using different antiscalants. Reduction of membrane product flow with time (**a**) and working pressure increase (**b**) using Aminat-K and Jurby-Soft.

**Figure 18 membranes-12-00852-f018:**
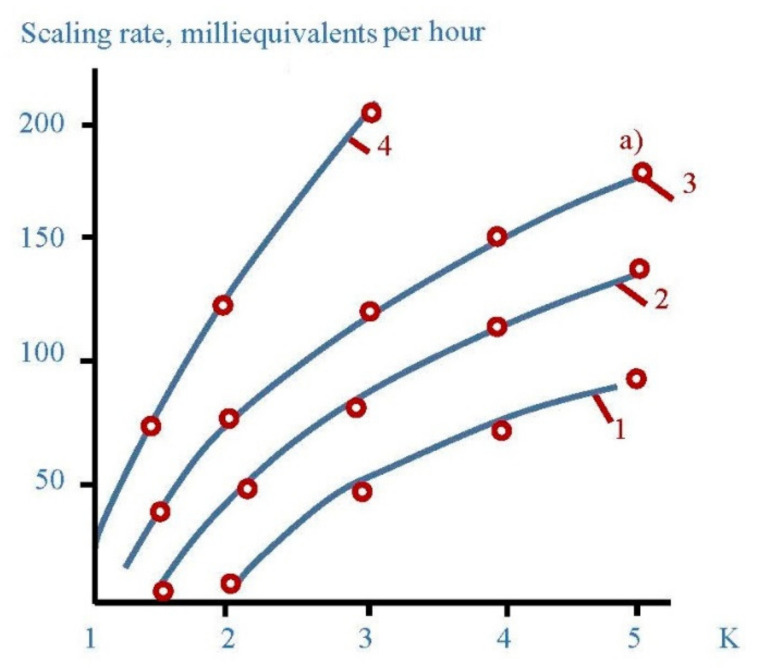
Prediction techniques for permeate flow reduction for the use of different antiscalants: plots (**a**) mass of scale versus time; (**b**) permeate flux reduction with time; (**с**) permeate specific conductivity growth with time.

**Figure 19 membranes-12-00852-f019:**
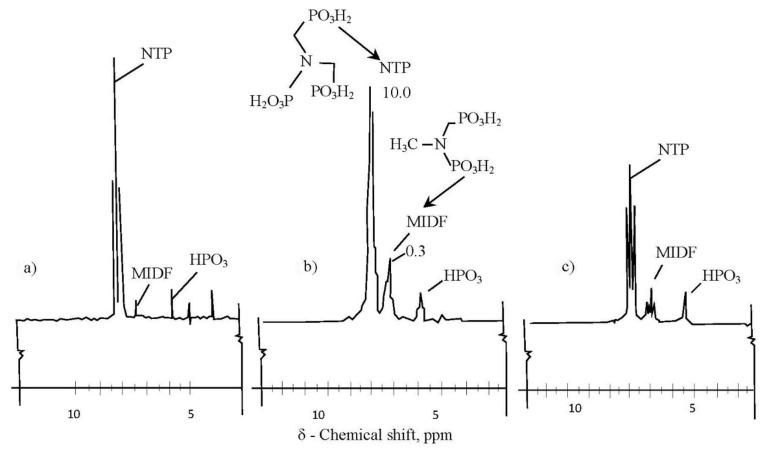
Spectrums of nuclear magnetic resonance investigations. The resonant adsorption frequencies versus chemical shift, (**a**) Jurby-Soft; (**b**) Aminat-K; (**c**) Actoscale.

**Table 1 membranes-12-00852-t001:** Determination of calcium carbonate supersaturation values, reached in “dead” areas at different recoveries (and values of coefficient K).

Number of Sample	K	Ca^-^	HCO_3_^-^	pH	TDS, ppm	K_sp_, Milliequiva-Lents^2^·Liter^−2^	$\frac{\left[CO_{3}\right]}{\left[HCO_{3}\right]}$	CO_3_	$\mathrm{Supersaturation}:\;\frac{\left[Ca\right]\left[CO_{3}\right]}{K_{sp}}$
1	1	4.5	2	7.3	200	5.22 × 10^−9^	0.01	0.02	20
2	2	8.0	3.2	7.6	380		0.015	0.04	100
3	3	12	6	7.9	520		0.018	0.1	200
4	4	18	7.4	8.0			0.02	0.14	540
5	5	20	9	8.2	650		0.03		1000
6	15	40	40	9.0	4600		0.3	12	1.2 × 10^5^
7	20	70	60	9.2	5700		0.4	24	3.3 × 10^5^
8	30	100	80	9.3	7300		0.5	40	7.8 × 10^5^

**Table 2 membranes-12-00852-t002:** Main characteristics of crystal growth.

Antiscalant dose	Dimensions[mm]	Volume [m^3^]	Crystal Mass [kg]	Number of Crystals	Calcium Carbonate Amount, Milliequivalents	Crystal Surface [m^2^]	Total Crystal Surface [m^2^]	Nucleatim Rate—Number of Nuclei per Hour
2	7	35 × 10^−17^	35 × 10^−14^	1.6 × 10^10^	5.6 × 10^−3^	30 × 10^−11^	4.8	0.8 × 10^10^
5	5	12.5 × 10^−17^	12.5 × 10^−14^	3.2 × 10^10^		15 × 10^−11^	5.8	1.6 × 10^10^
10	3	2.7 × 10^−17^	2.7 × 10^−14^	12 × 10^10^		5.4 × 10^−11^	6.5	6 × 10^10^

**Table 3 membranes-12-00852-t003:** Ratios of concentrations of the main components in phosphonicacid-based antiscalants.

No.	Antiscalant	NTF: MEDF	H_3_PO_3_ (per Cent of Total Phosphorous Content)
1	Aminat-K	5 : 1	3.0%
2	Actoscale	8 : 1	6.1%
3	Jurby-Soft	100 : 1	–

## Data Availability

Not applicable.

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
