# Peer review of "Investigation of Scaling and Inhibition Mechanisms in Reverse Osmosis Spiral Wound Elements"

_membranes, 2022, doi:10.3390/membranes12090852_

Round 1

Reviewer 1 Report

Reverse osmotic technique is an important approach in desalination and water treatment fields. The authors of this manuscript investigated the influences of crystal forming, nucleation behaviors and antiscalant adsorption on the membrane surface and pore structure. The results is helpful to understand the RO efficiency and life of RO membranes. Before publication, I still have some concerns about the influence mechanism. 

1.  That would be better if the authors can add the permeability-selectivity tradeoff with considering the dead areas.

2. The crystal formation is a time-dependent process, which will affect both the water flux and selectivity. So what does the author define the dead area ? for example, the region where water flux is zero ? 

3. There are also some important work to break the tradeoff between the permeability and selectivity [Science 2019, 364: 1033; Science Advances 2020, 6(34): eaba9471]. Please discuss them in the Introduction section.

Author Response

Author is intended to thank the Reviewer for patience and time to read the article and for making important professional comments. The author is glad to give explanations to the comments and questions that arose while reading the article.

Comment 1: That would be better if the authors can add the permeability-selectivity tradeoff with considering the dead areas.

Reply to the comment 1: In the text we tried to make explanation of dead areas behaviour. The described calcium carbonate growth mechanism was investigated throughout 40 years and was presented at a number of EDS and IDA conferences as well as published in Desalination and Desalination and water reuse magazines [6, 32-35]. To have better understanding of membrane rejection and permeability tradeoff, we suggest reading articles where more detailed description is provided.

We decided to add the following description to the text of the article:

"The dead area in membrane separation process is recognized as the membrane area with low cross-flow velocity of the separated solution, which causes an increase of concentration polarization and leads to a decrease of rejection and increase of permeate salinity. The increase of concentration polarization also leads to the increase of salt concentration at the membrane surface that causes supersaturation of sparingly soluble salts and their precipitation.  The level of concentration polarization does not grow indefinitely. The volume of liquid above the dead area of membrane surface follows the mass balance: the amount of salts that enters the dead area from the bulk solution is equal to the amount of salts withdrawn with permeate".

This mechanism is illustrated by the Figure 3. One of the main goals of the research presented in the article was evaluation of chemical composition of solution above the membrane surface in dead area. The experimental technique is illustrated by graphs on the Figures 7 and 8. We determine TDS and concentrations of water species in the feed water, and then we operate the test unit in circulation mode and increase TDS and concentration values in the circulated solution and determine TDS concentration values in the product samples withdrawn throughout the test cycle. The moment, when permeate TDS corresponds to the TDS of the feed water solution is the moment when behaviour of dead area reaches a mass balance. This technique was used to evaluate chemical composition of water solution (and supersaturation values) in dead areas in membrane modules with different membranes (membranes with different rejection characteristics).

Comment 2: The crystal formation is a time-dependent process, which will affect both the water flux and selectivity. So what does the author define the dead area? for example, the region where water flux is zero?

Reply to comment 2: The answer to the second question is described above

Comment 3: There are also some important work to break the tradeoff between the permeability and selectivity [Science 2019, 364: 1033; Science Advances 2020, 6(34): eaba9471]. Please discuss them in the Introduction section.

Reply to comment 3: Yes, we agree that this important problem should be discussed in details. Unfortunately, a large number of researches devoted to sparingly soluble salts formation on membrane surface (membrane scaling) lack evaluation of real conditions of crystal formation on membrane surface. We have mentioned a number of the serious researches in this field in references [7-10], and we add the additional comments to the Introduction section: "Rejection and permeability characteristics of membranes also influence the scaling propensities of membrane modules. It can be explained by the dead areas formation in on membrane surface and increase of concentration polarization ratios in these areas that tend to supersaturation and deposition of sparingly soluble salts. Rejection characteristics of membrane as well as their flux influence concentration polarization ratio and thus influence scaling rates. This was confirmed by a number of researches, where nanofiltration membrane spiral wound modules demonstrated lower scaling propensities than modules tailored with reverse osmosis membranes [7]".

Again, the author thanks the reviewer for the recommendations for improving the article.

Reviewer 2 Report

This paper studies experimentally the formation of scaling and how antiscalant reduce scaling. The paper is written satisfactorily; however, there are many scaling has been done especially in the presence of spacer. Can the author compare this work against other experimental study in the literature? The novelty of this works is more towards of how antiscalant reduces scaling, but not the understanding of scaling formation. 

Author Response

The author is ought to thank the reviewer for reading the article and making valuable professional comments.

Comments: This paper studies experimentally the formation of scaling and how antiscalant reduce scaling. The paper is written satisfactorily; however, there are many scaling has been done especially in the presence of spacer. Can the author compare this work against other experimental study in the literature? The novelty of this works is more towards of how antiscalant reduces scaling, but not the understanding of scaling formation.

Reply to the comments: The question raised by reviewer requires some clarification.

The described scaling mechanism as well as behaviour and role of antiscalants in preventing scaling was investigated by the author throughout 40 years and was presented in the form of reports ay EDS and IDA conferences as well as published in "Desalination" and "Desalination and water reuse" journals. The main goal of this article is to describe and present mechanism of scaling formation. It is well known that formation of crystals occurs in two stages: nucleation and crystal growth. Nucleation occurs when supersaturation exceeds solubility limits. To prevent scaling, antiscalants are used. But to experimentally evaluate antiscaling efficiencies of various antiscalants, we need to know the supersaturation values of solutions to provide true and correct results. Unfortunately, the majority of researches devoted to scale prevention select test conditions (supersaturation values) that do not correspond to real conditions in membrane modules and often provide inconsistent results. Therefore, in our research we evaluate scaling rates in industrial modules that possess dead areas where supersaturation reaches high values. The other goal of our research was to evaluate supersaturation values that are reached in dead areas in commercial membrane modules. This is the reason why description of dead areas formation takes up so much space.

The author is very grateful that reviewer touched on this topic.

The author has reread the article and saw that quite a lot of criticism was presented towards colleagues that investigate scaling mechanisms and antiscalant efficiencies. I think adding more criticism would be excessive.

Round 2

Reviewer 1 Report

There are also some important work to break the tradeoff between the permeability and selectivity [Science 2019, 364: 1033; Science Advances 2020, 6(34): eaba9471]. That would be better if they can comment on these work in the Introduction seciton.